# Synergy of the microRNA Ratio as a Promising Diagnosis Biomarker for Mucinous Borderline and Malignant Ovarian Tumors

**DOI:** 10.3390/ijms242116016

**Published:** 2023-11-06

**Authors:** Enora Dolivet, Léopold Gaichies, Corinne Jeanne, Céline Bazille, Mélanie Briand, Mégane Vernon, Florence Giffard, Frédéric Leprêtre, Laurent Poulain, Christophe Denoyelle, Nicolas Vigneron, Raffaèle Fauvet

**Affiliations:** 1ANTICIPE UMR (1086) (Interdisciplinary Research Unit for Cancers Prevention and Treatment), BioTICLA Laboratory (Precision Medicine in Ovarian Carcinoma), Federative Structure 4207 Normandie Oncologie, Université de Caen Normandie, Inserm, F-14000 Caen, France; l.gaichies@baclesse.unicancer.fr (L.G.); m.briand@baclesse.unicancer.fr (M.B.); m.vernon-contentin@baclesse.unicancer.fr (M.V.); f.giffard@baclesse.unicancer.fr (F.G.); l.poulain@baclesse.unicancer.fr (L.P.); c.denoyelle@baclesse.unicancer.fr (C.D.); n.vigneron@baclesse.unicancer.fr (N.V.); fauvet-r@chu-caen.fr (R.F.); 2Unicancer, Comprehensive Cancer Center F. Baclesse, 3 Avenue Général Harris, F-14000 Caen, France; c.jeanne@baclesse.unicancer.fr; 3Department of Pathology, Caen University Hospital, F-14000 Caen, France; bazille-c@chu-caen.fr; 4Unicancer, Comprehensive Cancer Center F. Baclesse, Biological Ressources Centre OvaRessouces, F-14000 Caen, France; 5Services Unit PLATON, Virtual’his Core Facility, Université de Caen Normandie, F-14000 Caen, France; 6CNRS, Inserm, CHU Lille, Institut Pasteur de Lille, US 41—UAR 2014—PLBS, University of Lille, F-59000 Lille, France; frederic.lepretre@univ-lille.fr; 7Unicancer, Comprehensive Cancer Center F. Baclesse, Calvados General Tumor Registry, F-14000 Caen, France; 8Department of Obstetrics and Reproductive Medicine, Université de Caen Normandie, F-14000 Caen, France

**Keywords:** microRNA, ratio, ovarian carcinoma, mucinous ovarian cancer, mucinous borderline ovarian tumor, cancer diagnosis

## Abstract

Epithelial ovarian cancers (EOCs) are a heterogeneous collection of malignancies, each with their own developmental origin, clinical behavior and molecular profile. With less than 5% of EOC cases, mucinous ovarian carcinoma is a rare form with a poor prognosis and a 5-year survival of 11% for advanced stages (III/IV). At the early stages, these malignant forms are clinically difficult to distinguish from borderline (15%) and benign (80%) forms with a better prognosis due to the large size and heterogeneity of mucinous tumors. Improving their diagnosis is therefore a challenge with regard to the risk of under-treating a malignant form or of unnecessarily undertaking radical surgical excision. The involvement of microRNAs (miRNAs) in tumor progression and their potential as biomarkers of diagnosis are becoming increasingly recognized. In this study, the comparison of miRNA microarray expression profiles between malignant and borderline tumor FFPE samples identified 10 down-regulated and 5 up-regulated malignant miRNAs, which were validated by individual RT-qPCR. To overcome normalization issues and to improve the accuracy of the results, a ratio analysis combining paired up-regulated and down-regulated miRNAs was performed. Although 21/50 miRNA expression ratios were significantly different between malignant and borderline tumor samples, any ratio could perfectly discriminate the two groups. However, a combination of 14 pairs of miRNA ratios (double ratio) showed high discriminatory potential, with 100% of accuracy in distinguishing malignant and borderline ovarian tumors, which suggests that miRNAs may hold significant clinical potential as a diagnostic tool. In summary, these ratio miRNA-based signatures may help to improve the precision of histological diagnosis, likely to provide a preoperative diagnosis in order to adapt surgical procedures.

## 1. Introduction

Ovarian cancers (OCs) are a heterogeneous collection of malignancies, together accounting for over 200,000 deaths per year worldwide [1]. One of the main factors contributing to the high death-to-incidence rate is the advanced stage of the disease at the time of diagnosis [2]. A late-stage presentation has a 5-year relative survival rate of 29%, in contrast with 92% for early-stage disease [3]. However, this description of OC is mainly driven by the most common OC type, high-grade serous OC (HGSOC), which represents approximately 75% of cases. Actually, OC is a collection of separate disease entities (histological subtypes or histotypes), each with distinct molecular landscapes, developmental origins and clinical behavior. To date, while the vast majority of research efforts have focused on the most common OC histotype leading to improvements in the understanding and treatment of HGSOC, the characteristics of rarer histological subtypes, such as mucinous ovarian carcinoma (MOC), remain poorly understood [4]. MOC is a rare tumor, accounting for ≤5% of OC cases,1 and is diagnosed in younger patients, with a median age at diagnosis of 50–54 years [5]. The prognosis for women with MOC, all stages combined, is higher than that of HSGOC with 58% (vs. 40%) 5 year-survival [6]. However, women with advanced stages (III–IV) of MOC have a three times lower survival than those with HGSOC, notably due to their poorer response to first-line platinum-based chemotherapy [7]. Mucinous histotype has also a different continuum of malignant progression between a benign, borderline and malignant tumor. Mucinous malignant forms represent a small proportion of ovarian tumors, alongside borderline and benign components, representing 15% and 80%, respectively [8]. In this context, borderline mucinous tumors are clinically distinguished from malignant tumors by their earlier onset of 10 years (41 years on average), their earlier diagnosis (90% at stage I) and their better survival (95% survival at 5 years) [9,10]. The therapeutic management of MOC relies on the debulking surgery, whose extension is adapted to the FIGO stage of the tumor, with the objective of complete resection [5]. Its radicality is also adapted to the characteristics of the tumor, with bilateral adnexectomy, hysterectomy, appendectomy, omentectomy and lymph node dissections for malignant tumors or a conservative surgery limited to the tumor ovary for borderline tumors in order to preserve the fertility of these women. For malignant tumors, conventional adjuvant platinum-based chemotherapy is recommended after excision, whereas it is not indicated for borderline tumors, even in the presence of microinvasion or intraepithelial carcinoma [11].

In the clinic, because of the size and the heterogeneity of mucinous tumors, malignant forms are difficult to distinguish from the borderline and benign subtypes, which have a better prognosis. Thus, during the surgical exploration, an extemporaneous examination is carried out in order to adapt the extent of the excision, but it also lacks precision and only agrees with the definitive histology in 64.7% of cases, frequently underestimating the invasive forms [12]. The challenge of managing mucinous tumors is therefore based on surgical treatment adapted to the most exact histological diagnosis possible [13,14]. Thus, the pre- or intraoperative distinction of the borderline or malignant nature of mucinous tumors would constitute a progress for their surgical management, the quality of which is a major prognostic factor for malignant tumors and the extent of which conditions patients’ pregnancy plans. In this context, microRNAs (miRNAs) may constitute tumor and/or circulating molecular signatures in order to improve the precision of histological diagnosis, likely to provide a preoperative diagnosis in order to adapt surgical procedures.

MiRNAs are small non-coding RNAs involved in the post-transcriptional regulation of gene expression [15]. MiRNAs play an important role in all biological processes, such as development, proliferation and cell death. The deregulation of miRNA function is associated with cancer development and progression, including OCs [16]. Interestingly, miRNA signatures can discern between normal ovarian tissue and OCs but are also able to discriminate between ovarian carcinoma histotypes (serous, endometrioid, clear cell and mucinous), underlying that cellular and/circulating miRNAs could be used as diagnostic and prognostic biomarkers in OCs. While many studies have looked for miRNAs as potential biomarkers of serous ovarian cancers, relatively few have focused on mucinous ovarian tumors. MiR-153 and miR-485-5p were found up-regulated in mucinous ovarian tumors relative to other ovarian histotypes but without any stratification on the malignant character [17]. MiR-192 and miR-194 and miR-215 were found associated with mucinous carcinoma [18,19]. Archived formalin-fixed paraffin-embedded (FFPE) specimens represent highly available resources for biomarker discovery, all the more so for this rare histological subtype. Interestingly, studies have shown good correlations between miRNAs quantified in paired fresh frozen and FFPE tumor samples independent of the methodology employed [20,21,22].

Improving the diagnosis of mucinous tumor is therefore a challenge with regard to the risk of under-treating a malignant form or of unnecessarily undertaking radical surgical excision. The aim of our study is to identify a diagnostic miRNA-based signature with the perspective of these new biomarkers potentially helping pathologists to better discriminate between malignant and borderline mucinous ovarian tumors and to improve personalized treatment. 

## 2. Results

### 2.1. Identification of Differentially Expressed miRNAs between Borderline and Malignant Ovarian Tumors by miRNA Profiling

MiRNAs were profiled using miRNA microarray in three borderline and three malignant ovarian tumors using FFPE patient samples (Table 1). 

Differentially expressed miRNAs were selected by two complementary approaches (Figure 1a and Appendix A). First, the selection was based on stringent criteria, which used a threshold of expression greater than 5 (in log2) and an expression variation of a factor greater than or equal to 2. This first step allowed to identify eight down-regulated miRNAs but only one up-regulated miRNA in malignant tumors compared to borderline mucinous tumors. Then, the second step was based on the complementary criteria of selection focused on over-expressed miRNAs and used only a fold-change greater than 2, which allowed us to identify 18 additional miRNAs of interest. Overall, this methodology aimed to identify 19 up-regulated and 8 down-regulated miRNAs between malignant and borderline ovarian mucinous tumors.

Among the 19 up-regulated miRNAs, 6 were classified as perfectly borderline and malignant tumors, of which 3 had an available PCR probe; and 9 were classified as having at most 2 errors and had an expression higher than 5 (in log2) for borderline tumor or higher than 20 (in log2) for malignant tumor, of which 8 had an available PCR probe. Among the eight down-regulated miRNAs, seven had an available PCR probe. Altogether, this study allowed us to identify 11 up-regulated miRNAs and 7 down-regulated between mucinous malignant and borderline tumors (Figure 1b). 

Otherwise, based on the results of miRNA profiling, we sought to identify normalizers for the subsequent analyses of miRNA expression. A panel of miRNAs with a stable expression was identified using the NormFinder algorithm, from the expression profiles of the six tumors, as potential normalizers for the analysis of RT-qPCR results [23]. Among the normalizer candidates presenting an expression ≥ 5 (log2), a stability value lower than 0.25 and an available PCR probe, miR-4449, miR-4787-5p and miR-1915-3p were the three miRNAs with the highest stability and were retained for subsequent analyses (Appendix A). Significantly, the trend and the magnitude of variation in the 18 miRNAs of interest were not affected by the normalization using the geometrical mean of the three normalizers (Figure 1b).

### 2.2. Validation of Differentially Expressed miRNAs between Borderline and Malignant Ovarian Tumors by RT-qPCR

All miRNAs selected were analyzed by RT-qPCR, the gold standard method, with miRNA-specific probes for the validation of the results, obtained with the microarray approach. First, the stability of the expression of the three normalizers was similar between the two approaches (Appendix A), allowing us to use them for further RT-qPCR experiments. Using the same six previous samples, all the down-regulated miRNAs (7/7) and more than 50% of up-regulated miRNAs (6/11) identified by microarray were validated by RT-qPCR experiments. Of note, for the other up-regulated miRNAs, an inversion of direction of the trend of variation (miR-1202, miR-6132 and miR-3136-3p) was observed, and no difference was found for miR-6126 and miR-4668-5p (Appendix A). 

The differential expression of these 13 miRNAs validated by RT-qPCR in the discovery set was next assessed using the validation set, in 11 borderline tumor and 12 malignant tumor samples (1 malignant tumor sample was excluded because the RNA concentration was too low) (Figure 2). As described previously for the discovery set, the seven down-regulated miRNAs were validated on the entire cohort (miR-195-5p, miR-152-3p, miR-487b-3p, miR-497-5p, miR-199a-3p, miR-130a-3p and miR-26a-5p). Moreover, five of the six up-regulated miRNAs were also validated on the entire cohort (miR-3175, miR-210-3p, miR-375, miR-4443 and miR-182-5p). Of note, only one miRNA (miR-4443) was significantly over-expressed in the malignant tumor group (*p* = 0.008). 

Interestingly, the 3 miRNAs (miR-1202, miR-6132 and miR-3136-3p) showing an inversion of the direction of variation in the discovery set also presented a similar variation in the entire cohort, suggesting that they could be used as down-regulated miRNAs to obtain a total of 10 under-expressed miRNAs in malignant tumors. Two over-expressed miRNAs were excluded because no differences were found between the two groups: miR-6126 and miR-3135b with a fold-change at −0.08 and −0.17, respectively. Altogether, after the RT-qPCR step, this study confirmed 5 up-regulated miRNAs and 10 down-regulated miRNAs between malignant and borderline mucinous tumors for further analyses. 

### 2.3. MiRNA Expression Ratio Allows to Distinguish Borderline and Malignant Tumors but Not Optimally

To overcome issues linked to data normalization and to improve the accuracy of their diagnostic value, we used a ratio analysis, which consists of measuring and comparing the expression ratio of up-regulated to down-regulated miRNAs. Then, 50 ratios were calculated per samples by dividing the expression value of each of the five over-expressed miRNAs in malignant tumors by the expression value of each of the 10 under-expressed miRNAs in malignant tumors (Figure 3). Twenty-one ratios were significantly different between malignant and borderline tumor samples. However, no ratio was able to perfectly discriminate between the two groups (Appendix A). The sensibility and the specificity of the miRNA ratio were between 33% and 100% and 54% and 100%, respectively. The best area under the curve was observed for the miR-182-5p/miR-199a-3p expression ratio with an AUC of 0.90, a sensibility of 67% and a specificity of 100%.

### 2.4. Pairs of Ratios of miRNAs Demonstrated High Accuracy to Discriminate between Borderline and Malignant Tumors

In order to increase the efficiency of this methodology based on the miRNA expression ratio, we combined the ratio two by two using the threshold defined by ROC curves and Youden index for each sample. Importantly, pairs of ratios based on less than four distinct miRNAs were removed from the analyses. These results are shown as a heat map of the number of classification errors (Figure 4). 

Out of the 900 combinations tested, 14 (1.56%) double miRNA ratios classified all the samples perfectly (Figure 5). These combinations of miRNA expression ratio involved 2 up-regulated miRNAs (miR-210-3p and miR-375) and 9 of the 10 down-regulated miRNAs (except miR-6132), suggesting that these double miRNA-based ratio signatures could be used to discriminate between mucinous malignant and borderline ovarian tumors.

## 3. Discussion

The development of new diagnostic tools for the identification of malignant mucinous tumors of the ovary could improve the management of aggressive forms and preserve the fertility of women with borderline forms. In this context, miRNAs could constitute diagnostic signatures and provide additional information to that of biopathological examinations to offer the most reliable diagnosis possible to patients and to best guide clinicians’ therapeutic decisions. 

To date, the pathological examination of a mucinous tumor is a difficult exercise due to their large size and their heterogeneity. No radiologic or biologic exam allows for a certain diagnosis in preoperative stages. Therefore, sometimes, it is carried out an extemporaneous analysis during the intervention to guide the therapeutic procedure. The performance of this analysis is limited given the limited time for its performance on complex and bulky tumors. Thus, a recent study conducted on 212 extemporaneous analyses of ovarian tumors of all histology showed a concordance between the extemporaneous diagnosis of borderline tumor and the definitive analysis of 60.6% [24]. The study showed an under-evaluation of invasive forms; 6.1% of tumors were later diagnosed as adenocarcinoma. Overall, mucinous tumors were the most frequently under-evaluated by extemporaneous analyses. A meta-analysis assessed the concordance between extemporaneous analyses and definitive examinations at 71.9%, with a sensitivity of 71.5% for a positive predictive value of 84.3%. Tumor size was the major criterion to explain these discrepancies [25].

The search for diagnostic tools for mucinous tumors of the ovary is difficult by their relative rarity; for example, our retrospective cohort recruited from two centers over a 10-year period. This may explain why numerous studies have focused on the expression profile of miRNAs either in the different histological subtypes of ovarian carcinoma or in the benign, borderline or malignant nature of tumors, but not specifically in mucinous tumors. One study focused on early-stage epithelial ovarian cancers, including mucinous histology (34 mucinous tumors/183 samples), and identified miR-192 and miR-194 as being associated with mucinous character [18]. Those two miRNAs miR-192 and miR-194, and miR-215 were also found up-regulated in ovarian mucinous carcinoma in a cohort of 73 ovarian tumors (19 malignant mucinous tumors) [19]. Also, miR-153 and miR-485-5p were identified as associated with a mucinous character in a cohort of 103 ovarian tumors that included 41 mucinous tumors (11 malignant, 20 borderline and 10 benign) [17]. Another study based on 149 FFPE samples of ovarian tumors (109 malignant, 23 borderline and 17 benign, including 24, 15 and 10 mucinous tumors, respectively) demonstrated that the expression of four miRNAs (miR-30a-3p, miR-30c, miR-30d and miR-30e-3p) was decreased in mucinous tumors compared to other different ovarian carcinoma histological types [26]. To the best of our knowledge, this is the first report to ascribe differentially expressed miRNAs between mucinous borderline and malignant ovarian cancers, which have to be validated in a larger number of patients in further independent cohorts. However, the stringent selection of our cohort made it possible to improve its internal validity at the cost of a smaller number of patients. The internal validity of our cohort was reinforced by a double biopathological review, carried out by an expert, which made it possible to reclassify three borderline tumors as malignant and one malignant tumor as borderline. This last point underlines the difficulty to correctly diagnose borderline and malignant forms. This is even more the case for mucinous tumors, whose large size and heterogeneity require exhaustive sampling in order not to overlook an invasion territory.

Conversely, most studies are based on heterogeneous experimental and analytical conditions, particularly in terms of normalization, which is nevertheless a key stage in miRNA expression measurement. For example, in the above-mentioned studies, data were normalized with different strategies based on small-nuclear RNAs (U6, RNU48) or other housekeeping controls. In our study, for the first time in mucinous tumors, we identified three miRNAs (miR-4449, miR-4787-5p and miR-1915-3p) that could be used as endogenous normalizers for RT-qPCR experiments, even if they have to be validated on more patient samples. To find out more, we also used miRNA ratios that provide a guarantee of being unbiased by experimental conditions. 

In the present study, we analyzed the miRNA profiles of mucinous ovarian malignant tumors compared to those of borderline tumors using discovery and validation steps, leading to a panel of 18 miRNA candidates. Twelve out of eighteen miRNAs were validated by RT-qPCR across the entire cohort, i.e., a rate of 67%. Indeed, three miRNAs identified as over-expressed by miRNA profiling were finally validated as under-expressed and two miRNAs were not differentially expressed. These observations underline the limits of global approaches in terms of accuracy of results and the risk of false positives, and justify the use of individual validations by RT-qPCR after any global approach in order to control the results in a more sensitive and reproducible way. Furthermore, this validation rate highlights the efficiency of studies based on discovery and validation steps, even with a small sample size during the discovery phase. From a methodological point of view, these results illustrate the feasibility and relevance of a global approach on limited numbers and underline the role of miRNA selection criteria and, in particular, the relevance of their significance. The accuracy of these results could be improved by performing global approaches on more patient samples, but could also be explained by non-optimal normalization. It should be noted that the microarray chips are normalized in a standardized RMA (Robust Multichip Average) manner by several successive steps, including a subdivision of miRNAs by expression quantiles and taking into account the overall expression of each chip [27]. Moreover, this strategy assumes that few miRNAs are differentially expressed compared to the whole, and that there is a balance in the distribution of over- and under-expressed markers. As this normalization acts like a “black box” and has no equivalent for the normalization of individual validations, it exposes the experimenter to the choice of an unsuitable normalizer, which, by introducing variability, would reduce the validation rate of the candidate biomarkers. During this study, the choice of endogenous normalizers was based on the selection of “stable” miRNAs on the expression profiles. The selection of such stable miRNAs was carried out using the Normfinder algorithm [23]. If their stability seemed almost perfect in the hybridization chips, it is more akin to small insignificant variations in RT-qPCR, although carried out from the same RNA extracts. Moreover, their expression levels on the chips were not exactly correlated with those measured by RT-qPCR. Consistent with the observation of the lesser sensitivity of global approaches, these results underline the fragility of this strategy, which is nevertheless frequently used, consisting in selecting normalizers using global approaches to apply them during individual validations.

These recurring normalization biases in the study of miRNAs as biomarkers have led our team to combine the miRNAs of interest in the form of a ratio of an over-expressed miRNA to an under-expressed one. With the normalization factors thus canceling each other out, their potential biases disappear and the biomarker potential of the two miRNAs combines each other. Interestingly, by combining five up-regulated miRNAs and 10 down-regulated miRNAs, we produced 50 miRNA ratios of which 21 were significantly differentially expressed between malignant tumors and borderline tumors, whereas most of those miRNAs alone were not. In agreement with the literature [28,29], we also observed that the strategy based on the miRNA expression ratio improves the diagnostic potential of miRNA-based signatures. Some studies have also studied panels of miRNAs, or ratios, by combining them in different ways in order to raise their diagnostic or predictive accuracy. For example, a multiple biomarkers panel of five miRNA ratios, which had potential diagnostic value for esophageal adenocarcinoma, demonstrated enhanced specificity and sensitivity over a single miRNA ratio [30]. In our study based on the same approach, all ratios could perfectly discriminate between the two groups. However, in an original way, we showed for the first time that it was possible to combine miRNA expression ratios two by two in order to compensate their weaknesses and thus obtain a perfect classification of patients for 14 double ratios, with 100% sensitivity and 100% specificity. These observations are preliminary but are also encouraging to pursue this strategy on more sample patients, because it could open a new way to combine two miRNA expression ratios. 

We identified miRNA signatures discriminating between malignant and borderline mucinous tumors by working on the tumor areas that best characterize them. This approach is justified by the heterogeneity that characterizes these large tumors. However, it would be interesting to assess the diagnostic performance of these miRNA signatures in other territories. Therefore, a conservation of diagnostic performance could simplify the transfer of such signatures to clinical routine while ensuring a certain reliability. On the contrary, performance degradation would underline the specificity of these signatures and position them as precision tools. Particularly in tumors presumed to be borderline, the possible cohabitation of malignant and borderline territories confers a crucial role to sampling. These analyses on several territories could also provide a better understanding of the continuum between benign, borderline and malignant territories. However, due to the large number of signatures identified, the evaluation of their behavior with regard to tumor heterogeneity could follow the validation of some of the most efficient signatures in a suitable cohort. Ideally, the latter would be based on a multicenter prospective collection of tumor samples of malignant and borderline mucinous tumors validated by a double biopathological review. Regardless, the diagnosis made by the pathologist remains the gold standard and these miRNA signatures could constitute complementary tools to help them, especially in difficult cases.

At present, it is impossible to confirm the diagnosis preoperatively, and the clinician is only helped by imaging and the circulating markers CA 125 and CA 19.9. It could be interesting to study our miRNA signature in blood samples and to evaluate its relevance in the diagnostic orientation of a multicenter prospective collection of serum. Indeed, in certain studies, circulating miRNAs have already shown their diagnostic effectiveness within signatures, making it possible to discriminate malignant from benign and borderline ovarian tumors [16,31,32]. The limit of the liquid biopsy in mucinous tumor is that they are very large and heterogenous tumor whose malignant character can sometimes be exposed only in a small territory, which may not be sufficiently represented in the expression of circulating miRNAs. In addition, these tumors can sometimes also be associated with other histological types, the most common being Brenner tumors, with up to 33%, although this percentage is lower in malignant tumors than borderline, which can make the interpretation of miRNA expression profiles even more difficult [33,34].

In summary, the use of miRNA ratios on the most characteristic territory identified by the pathologist seems to be an easily applicable method with potential for general clinical use. The identification of such miRNA-based signatures, although they will require validation with a larger number of patient samples, may improve the diagnosis of mucinous tumors with regard to the risk of under-treating a malignant form or of unnecessarily undertaking radical surgical excision. These biomarkers could potentially help pathologists, whose final diagnosis remains the gold standard, to better discriminate malignant and borderline mucinous ovarian tumors, especially in difficult cases within the framework of precision medicine. 

## 4. Materials and Methods

### 4.1. Clinical Samples

Formalin-fixed paraffin-embedded (FFPE) tissue blocks were processed using surgically resected tumors. These samples and the associated data were obtained from patients who were followed at the Comprehensive Cancer Center F. Baclesse and the University Hospital (CHU) (Caen) from 1 January 2005 to 3 January 2017 and stored in the Biological Ressources Centre OvaRessouces and Tumorothéque de Caen Basse-Normandie (NF-S 96900 quality management, AFNOR No. 2016: 72860.5), respectively. All biological collections were declared to and authorized for transfer by the MESR (Ministry of Education, Health and Research, France, No. DC-2010-1243 or DC-2020-4333 and AC-2013-1852 or AC-2018-3356). We obtained written informed consent for patients still alive. Samples for which patients were lost to follow-up or for which informed consent could not be signed before death were the subject of a request for exemption and approved by the North-West Ethics Committee. Patient selection was conducted with the thesaurus of the Biopathology laboratory ADICAP (Association for the Development of Computer Science in Cytology and AnatomoPathology). The inclusion criterion in our cohort was an ADICAP code corresponding to mucinous borderline and malignant ovarian tumors (G4B4; G4B2; A4C4; G7B2). Among the 33 preserved tumors, 20 were mucinous borderline ovarian tumors and 13 were mucinous adenocarcinomas. The HES slides corresponding to the tumors of each patient were re-read by 2 pathologists (Dr C. Jeanne, Baclesse, and Dr C. Bazille, University Hospital) to confirm the histological diagnosis (2014 WHO classification) and identify their areas of interest (borderline and malignant). In accordance with the 2014 WHO classification, tumors presenting a borderline territory of less than 10% of the total surface area of the tumor were reclassified as benign tumors and excluded from the study. After re-reading 5 tumors presenting a borderline territory of less than 10% were reclassified as benign mucinous cystadenomas, three Borderline tumors were reclassified as malignant tumors and 1 malignant tumor was reclassified as borderline. Two cases (one borderline and one malignant) were excluded in the absence of available FFPE blocks and one malignant tumor was excluded due to its cellularity being lower than 50%. In summary, 11 samples of borderline tumors and 13 of malignant tumors were selected for this study (Appendix A). The clinicopathological characteristics of this cohort are summarized in Table 1. 

### 4.2. RNA Extraction

RNA from the FFPE samples was extracted using Nucleospin^®^ total RNA FFPE Kit (Macherey-Nagel, Hoert, France). We estimated the amount of tissue of FFPE nodes according to their surface (R*r*π) and the equivalence provided by the fabricant: 1 slice of 10 µm in thickness and 1 cm^2^ equates to 1 mg of material. We calculated that 20 µm thick slices were necessary to obtain 17–19 mg of material from each node. Slices were cut by microtome and the excess of paraffin was removed with a scalpel. According to the manufacturer’s protocol, the slices were incubated 5 min at 56 °C in the paraffin dissolver provided in the kit. The deparaffined samples were next incubated with proteinase K for 3 h at 56 °C to digest the fixed tissue, release acid nucleic acids and remove the crosslinks. Optimal binding conditions for small RNAs were adjusted and the lysate was applied to Nucleospin^®^ RNA column. Finally, the RNA was purified by washing steps and eluted. The purity and the concentration of RNA were assessed by absorbance spectrophotometry on a Nanodrop 2000 (Thermo Scientific, Les Ulis, France). One of the malignant tumors was excluded due to its low amount of RNA, which was insufficient for further analyses. 

### 4.3. MicroRNA Profiling by Microarrays

FFPE samples from three malignant and three borderline mucinous tumors were selected for microarray experiments. The concentrations ranged between 61.5 and 433.8 ng/μL. The A260/A280 ratio was between 1.8 and 1.9. Microarrays were processed according to the manufacturer’s recommendations. For each sample, 200 ng of RNAs were labeled with FlashTag Biotin HSR RNA Labeling Kit (Genisphere Affymetrix UK Ltd., High Wycombe, United Kingdom). Samples were hybridized for 16 h at 48 °C on GeneChip microRNA 4.0 Array (Affymetrix) and scanned with a GC30007G scanner (Affymetrix). Raw data were normalized using the Expression Console (Affymetrix) with the RMA method, algorithmically based on microarray spike-in data. Data were deposited in the NCBI Gene Expression Omnibus database (GSE245725).

### 4.4. MiRNA Expression by RT-qPCR

MiRNAs were retro-transcribed using specific stem-loop primers and the MicroRNA Reverse Transcription Kit (ThermoFisher Scientific). Briefly, 5 µL of the isolated RNA was mixed with 10 µL of RT master mix after a two-fold dilution step. Triplicates with 1.33 mL of cDNA were mixed with 18.7 mL of qPCR master mix (Universal Master Mix II without UNG). Fluorescence and threshold baselines were measured using an Applied ABI Prism 7500 Fast PCR system with the 7500 Software v2.0.6 (Applied Biosystems, Waltham, MA, USA). MiRNA expression levels were assessed by the 2 −∆Cq method or by relative quantification by the 2 −∆∆Cq method according to the MIQE guidelines [35].

### 4.5. Statistics

Excel (Microsoft) and GraphPad Prism8 software were used to draw graphs and to run Student’s *t*-tests under the equally variances hypothesis. Sensitivity and specificity were calculated using ROC curves, and optimal thresholds were objectively determined using the highest Youden index, with the formula Youden index = sensitivity + specificity − 1. Histogram data are the means and standard error of the mean at 95%. A *p*-value of <0.05 was considered significant. 

## 5. Conclusions

This study is the first one to focus on a homogeneous cohort of patients to define a miRNA signatures that could discriminate between mucinous malignant and borderline ovarian tumors. After identifying several differentially expressed miRNAs between the two groups, this is also the first study to demonstrate the interest in combining paired miRNA expression ratios to enhance the specificity and sensitivity, instead of a single miRNA ratio, to improve diagnosis accuracy. These miRNA-based signatures may become an easily applicable method with potential for general clinical use and provide additional clues for clinicians for decision-making to improve the management of mucinous ovarian cancer. 

## Figures and Tables

**Figure 1 ijms-24-16016-f001:**
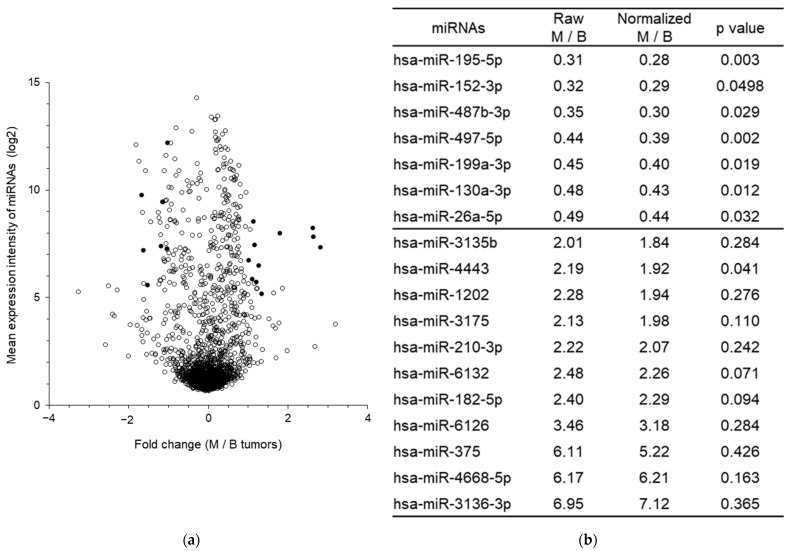
Selection of differentially expressed candidate miRNAs between malignant tumors (M) and borderline (B) mucinous ovarian tumors. (**a**) Volcano plot of the differentially expressed miRNAs as assessed by a microarray analysis in 3 malignant and 3 borderline ovarian tumor FFPE patient samples. The *y*-axis indicates the mean expression intensity of miRNAs (log2) and the *x*-axis is the fold-change (measured as the log2-transformed ratio of the mean expression between both groups). The black circles represent the 18 candidate miRNAs. (**b**) Fold-change of the raw and normalized miRNA expressions of the 18 candidate miRNAs between malignant and borderline tumors. The geometrical mean of miR-4449, miR-4787-5p and miR-1915-3p was used as the normalization method.

**Figure 2 ijms-24-16016-f002:**
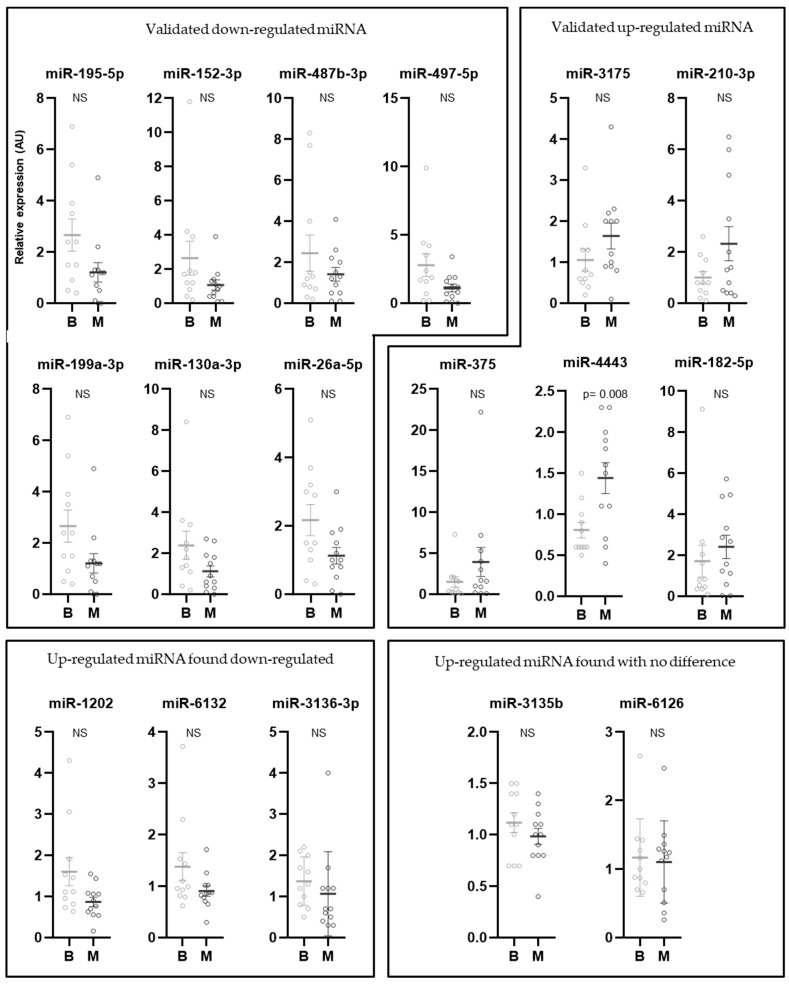
Validation of differentially expressed candidate miRNAs between borderline tumors (B) and malignant (M) mucinous ovarian tumors. MiRNA expression was analyzed by individual RT-qPCR in borderline (grey, N = 11) and malignant (black, N = 12) mucinous ovarian tumor FFPE patient samples. Results are presented as the normalized expression of candidate miRNAs between borderline and malignant mucinous ovarian tumors. The geometrical mean of miR-4449, miR-4787-5p and miR-1915-3p was used as the normalization method. NS: not significant.

**Figure 3 ijms-24-16016-f003:**
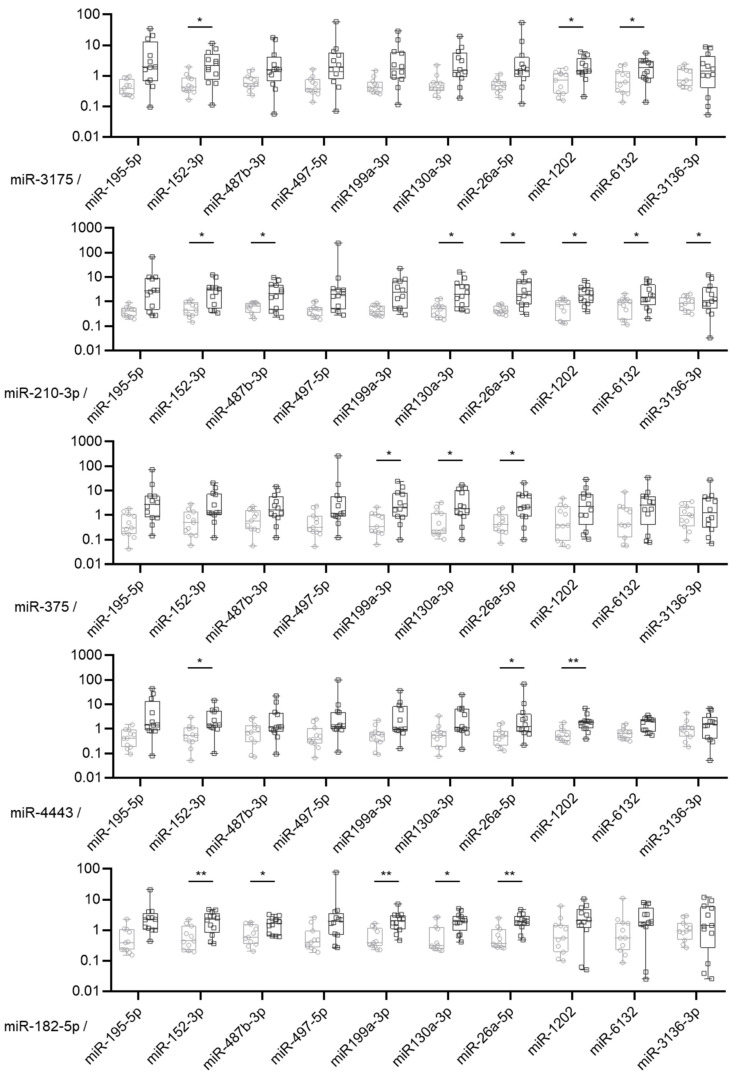
MiRNA expression ratio of 15 validated miRNAs differentially expressed between malignant and borderline tumors. Box plot of the relative expression of the ratio of the 5 up-regulated miRNAs on the 10 down-regulated miRNAs validated by RT-qPCR between borderline (grey, N = 11) and malignant (black, N = 12) mucinous ovarian tumors. * *p*-value < 0.05; ** *p*-value < 0.01.

**Figure 4 ijms-24-16016-f004:**
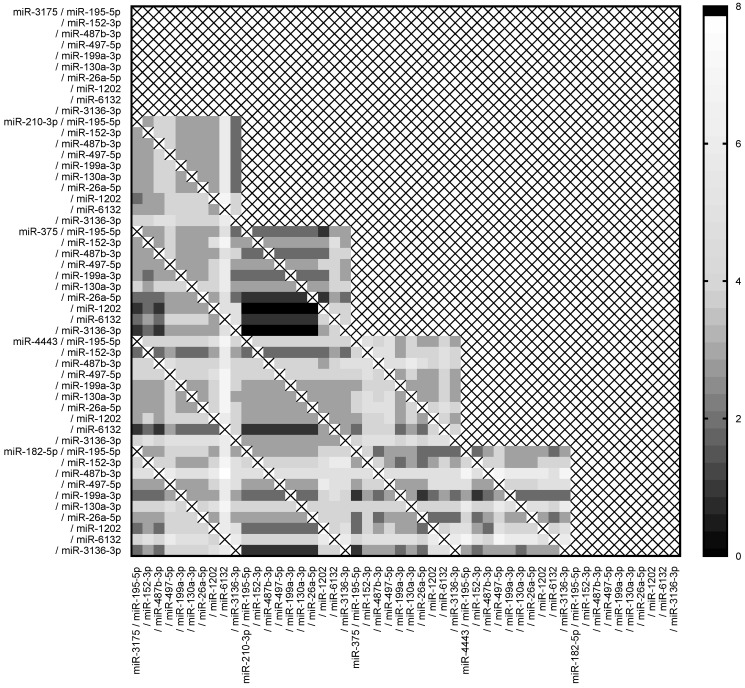
Combination of the miRNA expression ratio between malignant and borderline tumors. Heat map of the accuracy of pairs of miRNA ratios based on the number of errors to classify all borderline and malignant tumor patient samples. The threshold between borderline and malignant tumors was defined according to ROC curves and Youden index. Fourteen combinations of miRNA ratios have 100% accuracy.

**Figure 5 ijms-24-16016-f005:**
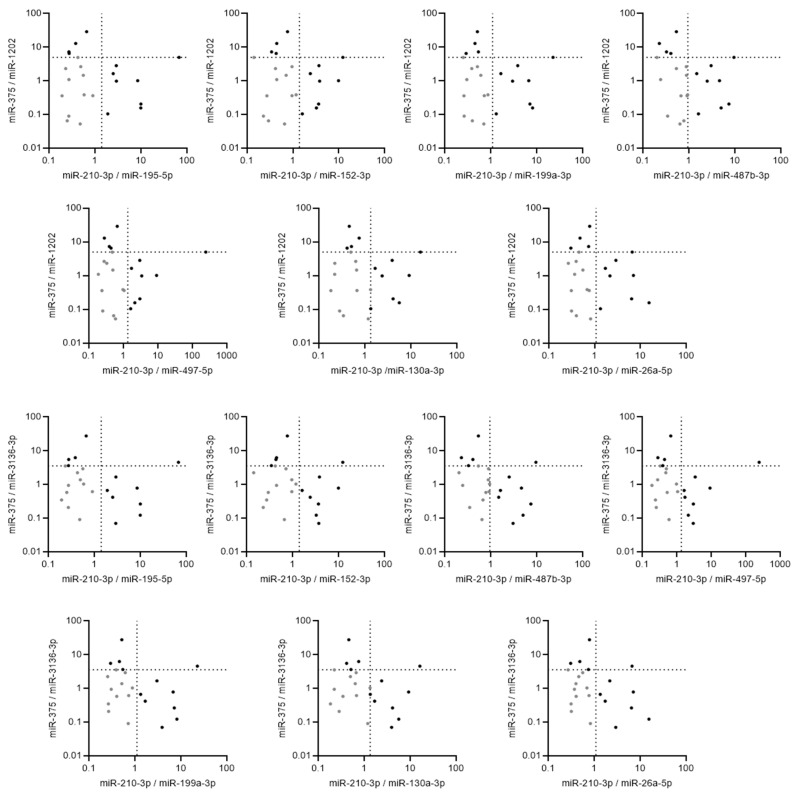
Graphical representation of the 14 pairs of miRNA ratios that are able to perfectly discriminate between borderline (grey dot) and malignant (black dot) mucinous ovarian tumors with 100% of accuracy. The threshold was defined according to ROC curves and Youden index and are shown as a dotted line for each of the 14 double miRNA ratios.

**Table 1 ijms-24-16016-t001:** Clinical features of patients according to the malignant or borderline character of the mucinous ovarian tumor.

	Malignant Tumor N = 13	Borderline Tumor N = 11
Age at diagnosis (years), Mean ± SD	46.9 ± 18.3	45.8 ± 20.7
BMI * (kg/m^2^), Mean ± SD	25.5 ± 5.0	23.8 ± 5.9
Menopaused, n (%)	4 (30.8)	5 (45.5)
Tumoral markers. Mean ± SD CA 125 (UI/mL) CA 19.9 (UI/mL)	59.6 ± 77.8 416.5 ± 806.8	35.6 ± 30.4 8.3 ± 4.2
FIGO ^#^ staging, n (%) I IV	11 (84.6) 2 (15.4)	11 (100) 0
Tumor size (cm), Mean ± SD	18.1 ± 7.5	17.8 ± 7.3

* BMI: Body Mass Index; ^#^ FIGO: International Federation of Gynecology and Obstetrics.

## Data Availability

Microarrays data were deposited in the NCBI Gene Expression Omnibus database (GSE245725).

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
