# Peer review of "Synergy of the microRNA Ratio as a Promising Diagnosis Biomarker for Mucinous Borderline and Malignant Ovarian Tumors"

_ijms, 2023, doi:10.3390/ijms242116016_

Round 1

Reviewer 1 Report

Comments and Suggestions for Authors

The manuscript “Synergy of microRNA ratio as promising diagnosis biomarkers for mucinous Borderline and Malignant ovarian tumorsby Enora Dolivet and co-authors to compare of miRNA microarray expression profiles between malignant and borderline tumors FFPE samples, identified 10 down-regulated and 5 up-regulated malignant miRNAs, which were validated by individual RT-qPCR. To overcome normalization issues and to improve the accuracy of the results, a ratio analysis combining paired up-regulated and down-regulated miRNAs was performed. Although 21/50 miRNAs expression ratio were significantly different between malignant and borderline tumor samples, any ratio could perfectly discriminate the two groups. However, a combination of 14 couples of miRNAs ratios showed high discriminatory potential, with 100% of accuracy in distinguishing malignant and borderline ovarian tumors, and suggested that miRNAs may hold significant clinical potential as a diagnostic tool. These ratio miRNAs-based signatures may help to improve the precision of histological diagnosis, likely to provide a preoperative diagnosis in order to adapt surgical procedures. However, there are concerns that must be taken into account before the work can be reconsidered for publication.

Comments

1.      It difficult isolate RNA from FFPE tissues. How to confirm the RNA quality?

2. The Gene Expression Omnibus (GEO) accession number of MicroRNA profiling by microarrays should be provided.

Comments on the Quality of English Language

 Minor editing of English language required.

Author Response

We thank you for your time and your comments which helped us improve significantly our manuscript. We tried to address your concerns to the best of our possibilities, as described below, and we hope that these modifications will find your appreciation. You will find below a point-by-point answer to all your comments. When changes were made in the manuscript according to such comments, we refer to the lines the changes are present in the updated version of the manuscript, so that changes can be more easily identified.

  1. FFPE material represents an extensive repository of tissue samples with a long-term clinical follow-up, providing a valuable resource for translational research. However, extracting good quality total RNA from FFPE samples is difficult due for example to the cross-linkage between nucleic acids and proteins, RNA modification as well as degradation during the fixation process and storage period.

In our study, RNA concentration and purity were assessed using spectrophotometer (Nanodrop). The concentrations of the FFPE samples used for miRNA profiling ranged between 61.5 and 433.8 ng/μl. The A260/A280 ratio was between 1.8 and 1.9. To date, RNA Integrity Number (RIN) is the most widely used approach to assess RNA degradation (Schroeder et al., 2006). In our study, the RNA quality was analyzed using microcapillary electrophoresis, which was performed on a 2100 Bioanalyzer (Agilent) with the “Eukaryote Total RNA Nano” assay. The average RIN was 2.2. This low RNA integrity number (RIN, around 2.0) is however typical for FFPE extractions. It is difficult to assess the RNA quality in FFPE samples. To date, there is a consensus on the fact, even though RIN is widely used for assessment of RNA quality, it is not a sensitive method to measure RNA quality in FFPE due to the possible RNA fragmentation (Chung, Cho, & Hewitt, 2016).

Otherwise, compared with longer RNA and DNA molecules, small RNAs, including miRNAs, are very robust and resistant to severe changes in pH or temperature, as well as the repeated ice/thaw cycles (Mitchell et al, 2008). Due to their small size, miRNAs may be also less prone to degradation and modification, so their analysis in FFPE specimens likely provides accurate results. Furthermore, good correlations with the results obtained using matched FFPE specimens and frozen samples have been reported in several miRNA profiling studies (Meng et al, 2013, Bucay et al, 2015; Vojtechova et al, 2017)). Overall, even though RNA degradation occurs, due to the formalin fixation process, these studies underline that it is feasible to extract miRNA from FFPE tissue and to perform quantitative analyses.

In our study, the choice of miRNA microarrays seems appropriate here, given the number of miRNAs detected and the high validation rate of individual miRNAs identified by RT-qPCR.  The recognized specificity of RT-qPCR suggests that what has been measured corresponds to the exact sequence of the miRNAs sought, and not to the nearby sequence that could be derived from a degradation product. In the case of random degradation, it would be possible to observe differences in expression on a small sample, but it is highly unlikely that they would be found on a larger sample. In our study, we observed concordant results for most of miRNAs during biological validation on all patients, suggesting that these were not random results resulting from mRNA degradation.

Our added information can be found lines: 113-117; 456-459 and 463-465

Schroeder, A.; Mueller, O.; Stocker, S.; Salowsky, R.; Leiber, M.; Gassmann, M.; Lightfoot, S.; Menzel, W.; Granzow, M.; Ragg, T. The RIN: An RNA Integrity Number for Assigning Integrity Values to RNA Measurements. BMC Mol. Biol. 2006, 7, 3, doi:10.1186/1471-2199-7-3.

Chung, J.-Y.; Cho, H.; Hewitt, S.M. The Paraffin-Embedded RNA Metric (PERM) for RNA Isolated from Formalin-Fixed, Paraffin-Embedded Tissue. BioTechniques 2016, 60, 239–244, doi:10.2144/000114415.

Mitchell, P.S.; Parkin, R.K.; Kroh, E.M.; Fritz, B.R.; Wyman, S.K.; Pogosova-Agadjanyan, E.L.; Peterson, A.; Noteboom, J.; O’Briant, K.C.; Allen, A.; et al. Circulating microRNAs as Stable Blood-Based Markers for Cancer Detection. Proc. Natl. Acad. Sci. U. S. A. 2008, 105, 10513–10518, doi:10.1073/pnas.0804549105

Bucay, N.; Shahryari, V.; Majid, S.; Yamamura, S.; Mitsui, Y.; Tabatabai, Z.L.; Greene, K.; Deng, G.; Dahiya, R.; Tanaka, Y.; et al. MiRNA Expression Analyses in Prostate Cancer Clinical Tissues. J. Vis. Exp. 2015, doi:10.3791/53123

Meng, W.; McElroy, J.P.; Volinia, S.; Palatini, J.; Warner, S.; Ayers, L.W.; Palanichamy, K.; Chakravarti, A.; Lautenschlaeger, T. Comparison of MicroRNA Deep Sequencing of Matched Formalin-Fixed Paraffin-Embedded and Fresh Frozen Cancer Tissues. PLOS ONE 2013, 8, doi:10.1371/journal.pone.0064393

Vojtechova, Z.; Zavadil, J.; Klozar, J.; Grega, M.; Tachezy, R. Comparison of the MiRNA Expression Profiles in Fresh Frozen and Formalin-Fixed Paraffin-Embedded Tonsillar Tumors., doi:10.1371/journal.pone.0179645

  1. The raw miRNA profiling data were uploaded to the NCBI’s Gene Expression Omnibus and are accessible through GEO Series accession number GSE245725. (https://www.ncbi.nlm.nih.gov/geo/query/acc.cgi?acc=GSE245725).

Our added information can be found lines : 470-471

Reviewer 2 Report

Comments and Suggestions for Authors

Excellent work. 

One question - do we have data regarding the change of miR expression during chemotherapy administration?

Author Response

We thank you for time and appreciation, we are glad that you enjoyed our manuscript. Guidelines, without sufficient literature, have not been issued regarding an indication for chemotherapy in advanced stages of Borderline tumor, even in the case of invasive implants. Adjuvant chemotherapy is not recommended, except in patients with high-risk clinical and pathological features. The National Comprehensive Cancer Network (NCCN) guidelines recommend surgery alone for stage IA or IB mucinous ovarian cancer and adjuvant platinum-based chemotherapy (carboplatin and paclitaxel, or oxaliplatin with fluorouracil or capecitabine) for more advanced disease. Likewise, the expansile type is associated with better prognosis and should not receive adjuvant chemotherapy, while the infiltrative form is associated with a high risk of relapse. In our cohort 6 malignant tumor out of 13 received chemotherapy (5 because of a FIGO stage higher than IB and 1 in front of an infiltrative form). All patients had a front line surgery and no tumors samples were available after chemotherapy. The analysis of miRNA expression during chemotherapy is very interesting, especially for study of predictive biomarkers. In our team, work is currently carried out on a cohort of serum collected at several stages of the treatment of patients with high-grade serous ovarian cancer (clinicaltrials.gov ID: NCT01391351).

Reviewer 3 Report

Comments and Suggestions for Authors

The Authors conducted a study on a homogeneous cohort of patients with mucinous ovarian tumors, with the aim to define a miRNAs signature to discriminate between Malignant and Borderline cases. The study is well conducted and well presented, even though there are some limits and it is still premature to apply these results in the clinical practice. The study, due to its originality and scientific interests, should be considered for publication, after some minor modifications.

I would suggest to insert a small paragraph, underlining the limits of this study, in particular explaining the limits regarding the application of this methodology in the clinical practice. The Authors should underline that currently the liquid biopsy has still many limits, and do not allow to substitute the role of anatomical pathologists, because the histological examination remains the gold standard for a precise diagnosis. However, the liquid biopsy may give important molecular information, possibly (not always) useful for some types of target therapies.

Finally, I would suggest to insert a little paragraph in the discussion where the Authors describe and cite what is currently known regarding miRNA in other ovarian cancers (doi: 10.1007/s43032-021-00717-w)  (doi: 10.1038/s41416-022-01925-0). It is also important to underline that very often ovarian cancer may present with mixed histotypes or as an evolution from a borderline to a malignant tumor and, rarely, may also change into other histotypes. An example that could be cited is in this article:  DOI: 10.3390/diagnostics11050827 .

Authors may speculate about the possible changes in miRNA signature in such kind of cases, presenting both mucinous and malignant components, along with evolution into other different histotypes.

Author Response

We thank you for your time and your comments, which helped us, improve significantly our manuscript. We tried to address your concerns to the best of our possibilities, as described below, and we hope that these modifications will find your appreciation. You will find below a point-by-point answer to all your comments. When changes were made in the manuscript according to such comments, we refer to the lines the changes are present in the updated version of the manuscript, so that changes can be more easily identified.

As you noted, our miRNA signatures cannot be currently used in clinic. The first step will be to validate our miRNA signature on a larger cohort of mucinous Borderline and Malignant ovarian tumors. We also need to analyze the expression of those miRNA on several territories of the same tumor to choice the most relevant signature. These steps are key points allowing us to validate our miRNA signatures. Whatever, the diagnosis made by the pathologist remains the gold standard and these miRNA signatures could constitute complementary tools to help them, especially in difficult cases.

About the liquid biopsy, as you cited, some studies analyzed circulating miRNA expression and have already shown their diagnostic effectiveness within signature making it possible to discriminate malignant from benign and Borderline ovarian tumors (Zhao et al, 2022; Gahlawat et al, 2022; Mandilaras et al, 2017). It could be interesting to evaluate whether our signature is also present in circulation and to evaluate its relevance in the diagnostic orientation on serum of patient with mucinous ovarian Borderline and malignant tumors. This miRNA signature could be associated with the imaging and the circulating biomarkers (CA 125 and CA 19.9) already used in practical clinic to helped the clinician before surgery although it is indeed the histological diagnosis which will allow the final diagnosis.

As you point out, the limit of the liquid biopsy in ovarian tumor is that they are often heterogeneous either by integrating different territories: benign, Borderline and malignant or by integrating, as you note to us, different histological types. Regarding mucinous tumors, they are most often associated with Brenner tumors as mesonephric like carcinoma or low grade serous component although this percentage is lower in malignant mucinous tumors than Borderline (Simons et al, 2019; Roma et al, 2015). This frequent heterogeneity actually makes the interpretation of miRNA expressions difficult in blood samples. The use of tumor or circulating miRNA signatures in heterogenic or mixed tumors requires similar studies to measure miRNA expression separately in each histological type.

Our modified section can be found in lines 381-383; 387-402 in the revised version of our manuscript.

Zhao, L.; Liang, X.; Wang, L.; Zhang, X. The Role of MiRNA in Ovarian Cancer: An Overview. Reprod. Sci. 2022, 10, 2760–2767, doi:10.1007/s43032-021-00717-w

Gahlawat, A.W.; Witte, T.; Haarhuis, L.; Schott, S. A Novel Circulating MiRNA Panel for Non-Invasive Ovarian Cancer Diagnosis and Prognosis. Br. J. Cancer 2022, doi:10.1038/s41416-022-01925-0

Mandilaras, V.; Vernon, M.; Meryet-Figuière, M.; Karakasis, K.; Lambert, B.; Poulain, L.; Oza, A.; Denoyelle, C.; Lheureux, S. Updates and Current Challenges in MicroRNA Research for Personalized Medicine in Ovarian Cancer. Expert Opin. Biol. Ther. 2017, 17, 927–943, doi:10.1080/14712598.2017.1340935

Simons, M.; Simmer, F.; Bulten, J.; Ligtenberg, M.J.; Hollema, H.; Van Vliet, S.; De Voer, R.M.; Kamping, E.J.; Van Essen, D.F.; Ylstra, B.; et al. Two Types of Primary Mucinous Ovarian Tumors Can Be Distinguished Based on Their Origin. Mod. Pathol. 2020, 33, 722–733, doi:10.1038/s41379-019-0401-y

Roma, A.A.; Masand, R.P. Different Staining Patterns of Ovarian Brenner Tumor and the Associated Mucinous Tumor. Ann. Diagn. Pathol. 2015, doi:10.1016/j.anndiagpath.2014.12.002
